# Parasite defensive limb movements enhance acoustic signal attraction in male little torrent frogs

**Longhui Zhao[1,2], Jichao Wang[2], Haodi Zhang[1], Tongliang Wang[2], Yue Yang[1], Yezhong Tang[1], Wouter Halfwerk[3], Jianguo Cui[1]***

[1]CAS Key Laboratory of Mountain Ecological Restoration and Bioresource Utilization & Ecological Restoration and Biodiversity Conservation Key Laboratory of Sichuan Province, Chengdu Institute of Biology, Chinese Academy of Sciences, Chengdu, China; [2]Ministry of Education Key Laboratory for Ecology of Tropical Islands, Key Laboratory of Tropical Animal and Plant Ecology of Hainan Province, College of Life Sciences, Hainan Normal University, Haikou, China; [3]Department of Ecological Sciences, Vrije Universiteit Amsterdam, De Boelelaan, Amsterdam, Netherlands

**\*For correspondence:**
cuijg@cib.ac.cn

**Competing interest:** The authors declare that no competing interests exist.

**Abstract** Many animals rely on complex signals that target multiple senses to attract mates and repel rivals. These multimodal displays can however also attract unintended receivers, which can be an important driver of signal complexity. Despite being taxonomically widespread, we often lack insight into how multimodal signals evolve from unimodal signals and in particular what roles unintended eavesdroppers play. Here, we assess whether the physical movements of parasite defense behavior increase the complexity and attractiveness of an acoustic sexual signal in the little torrent frog (*Amolops torrentis*). Calling males of this species often display limb movements in order to defend against blood-sucking parasites such as frog-biting midges that eavesdrop on their acoustic signal. Through mate choice tests we show that some of these midge-evoked movements influence female preference for acoustic signals. Our data suggest that midge-induced movements may be incorporated into a sexual display, targeting both hearing and vision in the intended receiver. Females may play an important role in incorporating these multiple components because they prefer signals which combine multiple modalities. Our results thus help to understand the relationship between natural and sexual selection pressure operating on signalers and how in turn this may influence multimodal signal evolution.

## Editor's evaluation

Zhao et al., present an intriguing proposal for the evolution of complex multimodal signals based on the analysis of both acoustic and visual signals of small torrent frogs' mating displays. Combining field observations with experiments, they suggest that male limb movements, which are used to swat away blood-sucking midges, have become attractive to female frogs, demonstrating how these movements enhance the mating calls of males. Their research suggests a potential pathway for hirtyle-host interactions to become co-opted into sexually selected mating displays.

## Introduction

Many animals can increase their communication efficiency by enhancing the complexity in a single sensory modality or by evolving displays that target multiple sensory modalities (*Cui et al., 2016*; *Partan and Marler, 1999*). Such multimodal signaling can be highly complex, often involving multiple

underlying neuronal motor programs that need to be synchronized in order to perform well (*Partan and Marler, 1999*; *Ryan et al., 2019*; *Halfwerk et al., 2019*; *Higham and Hebets, 2013*). The production and reception of multimodal signals is often more costly in terms of energy loss or predation and parasitism risk when compared to unimodal signals (*Bro-Jørgensen, 2010*), and their evolution is therefore often explained through functional benefits, such as cross-modal perception by receivers, which can improve signal detection and discrimination, or enhance attention and memory time (*Halfwerk et al., 2019*; *Hebets and Papaj, 2004*; *Halfwerk et al., 2014b*). Contra to *why*, we know far less *how* multimodal signals evolve from unimodal ones. An important question remains whether and when multimodal signals evolve de novo, or evolve through a process of co-option, by incorporating additional cues into a unimodal display (*Halfwerk et al., 2019*).

Human and many animals often generate by-product cues during signaling that can influence the perception of the dominant part of the signal. A famous example in humans is the McGurk effect which shows lip movement (the by-product of pronunciation) affects speech perception (*McGurk and MacDonald, 1976*). Similarly, floating frogs produce water ripples when calling from the water. These ripple cues are an unintentional by-product of calling, but have become part of the sexual display, as their presence modulates receiver responses to their acoustic signal components (*Halfwerk et al., 2014a*). Multimodal signals can thus originate from cues associated with primary signal production, either through a physical linkage (e.g. case of call-induced water ripples) or through a temporal linkage, for example, cues generated by other non-communicative behaviors that occur around the time of signaling. Once signal receivers start to pay attention to these other cues, subsequent selection on these physically or temporally linked cues may lead to closer integration and synchronization with the primary signal and become incorporated into a new, multimodal display. Such process of co-option has been proposed for many ritualized visual displays which are predicted to have evolved from different intra- or interspecific activities such as intentional movements, protective and autonomic responses (*Harper, 1991*; *Hödl and Amézquita, 2001*). For example, comparative analyses on Anatidae (i.e. ducks) suggest that the precopulatory displays of head-dipping seem to be derived from bathing behavior (*Johnsgard, 1962*), which may have originally been associated in time with other sexual signaling behaviors. Physical movements are in particular likely to become integrated into sexual displays. In human and some animals, movements are often used to attract the attention of receivers (*Clark and Morjan, 2001*; *Hugill et al., 2010*) and may even serve as primary sexual displays providing reliable information on sender quality (*Hugill et al., 2010*; *Hasson, 1997*; *Taylor et al., 2000*).

Anurans (i.e. frogs and toads) provide a good opportunity to test whether non-communicative behaviors can be co-opted into a signal function (*Preininger et al., 2013a*; *Starnberger et al., 2014*). In some anurans, for example, arm-waving movements appear to originate from cleaning behavior due to the similarities in both displays (*Pombal et al., 1994*). Furthermore, many anuran species display defensive movements that are also similar with communicative visual displays of some taxa (*de Sá et al., 2016*; *Preininger et al., 2009*). Here, we hypothesized that some of these physical movements originated from anti-parasite behavior in frogs. Amphibians are often confronted with parasitism from a range of different insects, such as mosquitoes or midges (*Grafe et al., 2008*; *Legett et al., 2018*; *Van Beurden, 1980*; *Kay et al., 1985*; *Ferrar, 1987*). These parasites are often attracted to the frog's mating call to collect a blood meal. In return they may transmit blood-borne diseases or endo-parasites, thus imposing a large cost to a calling frog (*Bernal and de Silva, 2015*; *Bernal et al., 2006*; *Meuche et al., 2016*). Some anurans are observed to perform defensive physical movements in response to these parasites.

Anurans primarily communicate with acoustic signals. Background noise generated by flowing water is an impediment to acoustic communication of torrent anuran species (*Preininger et al., 2013a*; *Grafe et al., 2012*). Multimodal displays may increase the efficiency of signal detection and perception (*Hebets and Papaj, 2004*), and thus are favored in complex noisy environments. Researches on some torrent frogs have showed that the coupling between physical displays (e.g. vocal sac and limb movements) with calls may direct the receiver's attention and play an important role in conspecific interactions (*Preininger et al., 2013a*; *Preininger et al., 2009*; *Grafe et al., 2012*; *Grafe and Wanger, 2007*). Insect-evoked movements are often similar to limb movements that act as visual signal. The movements evoked by host-parasite interactions may increase the complexity of signal displays and can potentially increase female preference for acoustic signals in frogs. Thus, the

parasite-host interaction may provide an important source of evolutionary raw material for ritualized visual displays in anurans, which in turn may have led to the evolution of multimodal displays in which visual movements and acoustic calls have been combined.

In the present study we examined the role of insect parasites in driving multimodal signal evolution in the little torrent frog (*Amolops torrentis*), a tropical species that breeds in noisy mountain streams and displays both day and night. Male little torrent frogs prefer to emit advertisement calls from the rocks near streams or near vegetation. Calling males are also often observed to be disturbed by various insects including midges and some other potential parasites. In order to repel these insects, they usually display limb movements that are similar to some spontaneous movements as well as to visual displays that have been reported in other torrent frogs (*de Sá et al., 2016*; *Caldart et al., 2014*). Frog-biting midges often locate male anuran hosts by exploiting their advertisement calls (*Grafe et al., 2008*; *Bernal et al., 2006*; *Meuche et al., 2016*). So, there may be a predictable connection between stereotyped calls and limb movements that seem to be chaotic in this species. These contents suggest that their defensive movements may act as a visual cue component in addition to the acoustic component of little torrent frogs.

Here, we evaluated whether the presence of eavesdropping parasites increases limb movements and determined whether and how these parasite-evoked physical displays influence female preference for multimodal signals. First, we filmed little torrent frogs in the field and classified their physical displays involving limb movements. Second, we assessed the link between parasitism and male limb display by quantifying the frequency of parasite interactions as well as limb movements, and tested whether calling males produced more parasite-evoked displays compared to silent males. In order

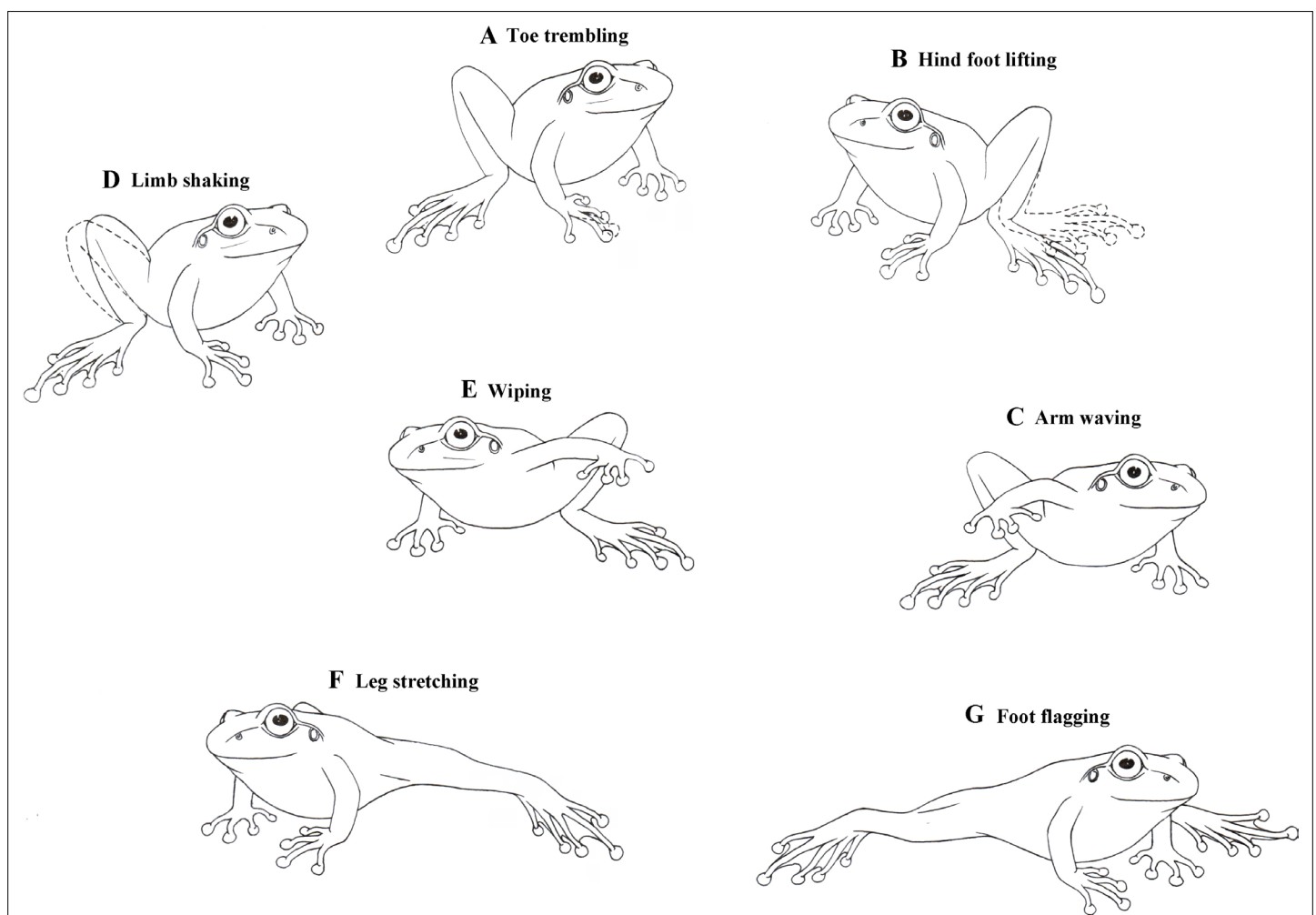

**Figure 1.** Diagrammatic drawings of seven limb motion displays shown by male little torrent frogs.

to avoid assessing individual differences, we collected sufficient samples (at least 30 males in each group) to compare the defend displays between calling frogs and silent frogs inhabited in same environment. Third, we determined the effect of midge-evoked visual movements on female mate choice with and without the advertisement calls presented, and examined if there was a specific sequence of attractive limb movements and calls. Finally, we tested whether exaggerated movements play a role in mate choice by analyzing the change of male foot flagging (FF) and leg stretching (LS) displays when females approached them in the field.

## Results

### The diverse repertoire of limb displays

Male little torrent frogs possess a rich repertoire of visual displays involving the movements of limbs. Their definitions and detailed descriptions (modified from previous reports; *Hödl and Amézquita, 2001*; *de Sá et al., 2016*) were as follows: (1) Toe trembling (TT): Vibrating, wiggling, or twitching the toes, with the arm and leg motionless (*Figure 1A*). (2) Hind foot lifting (HFL): Raising one hind foot towards the dorsal direction and then returning it back on the ground, without extending the leg (*Figure 1B*). (3) Arm waving (AW): Lifting one of two arms and waving it up and down in an arc toward the front of head (*Figure 1C*). (4) Limb shaking (LSA): Rapid movements of hand or foot in an up-and-down pattern (*Figure 1D*). (5) Wiping (W): Moving a hand or foot on the ground, with the limb not fully extended (*Figure 1E*). (6) Leg stretching (LS): Stretching one leg or both legs at the substrate level (*Figure 1F*). (7) Foot flagging (FF): Raising one or both legs off the substrate level, extending it/them out and back in an arc shape, and then getting it/them back to the ground (*Figure 1G*). Both LS and FF were movements of the hind limb and they were occasionally performed in a similar pattern that was not easy to distinguish. We therefore categorized these two movements collectively LS + FF throughout the remaining part of this study.

### Parasites induce more limb movements

Five observed limb movements are not only spontaneously generated, but also induced by insects (*Figure 2A*). The passive visual movements were predominantly evoked by some potential hematophagous parasites such as midges and sandflies (*Videos 1 and 2* and *Figure 3*). Specifically, we identified *Corethrella* spp. midges and *Phlebotomus* flies, which prefer to feed on the blood of ectotherms (*Bernal and de Silva, 2015*; *Kato et al., 2010*). As seen in *Figure 2A*, the movements that were produced by parasite interactions had a high proportion of visual cues, such as W (49.3%), AW (38.8%), LSA (32.9%), and HFL (20.4%), while the proportion was low in the TT (12.5%) category. LS + FF were not induced by parasite interactions (*Supplementary file 1*). Interestingly, we observed a positive correlation between the level of parasite interference and the total number of visual movements (Pearson's correlation; $N=39$; $R=0.830$; $p<0.001$; *Figure 2B*). We also ran the same analyses but restricted to the two movements found to be attractive to females (i.e., AW and HFL movements; see also below). As a result, we found the same correlation between AW/HFL and the presence of parasites (Pearson's correlation; $N=39$; $R=0.737$; $p<0.001$).

### Calling males show more parasite-evoked limb movements

We compared the limb movements between calling individuals ($N=39$) and non-calling individuals ($N=30$) in the presence and absence of eavesdropping insects. Among the six types of limb movements, the TT and LS + FF were rarely induced by parasitic insects (*Supplementary file 1*). We thus only compared the difference between calling males and silent males for the other four visual displays. We found calling males to produce more defensive AW (Wilcoxon rank sum test; $W=895$; $p<0.001$), W ($W=872.5$; $p<0.001$), LSA ($W=755$; $p=0.032$) and HFL ($W=736.5$; $p=0.053$) display than silent males (*Figure 4*), presumably because calling individuals attracted more parasites than silent individuals. We also compared the whole number of parasite-evoked movements between calling males and silent males. The result suggested that calling individuals have more parasite visits than non-calling individuals (Wilcoxon rank sum test; $W=900.5$; $p<0.001$).

### The role of parasite-evoked limb displays on female choice

Females expressed a strong preference during audio-visual playbacks to approach a male that performed an AW (probability = 0.81; $N=16$; $p=0.021$; *Figure 5A*) or HFL (probability = 0.76; $N=21$;

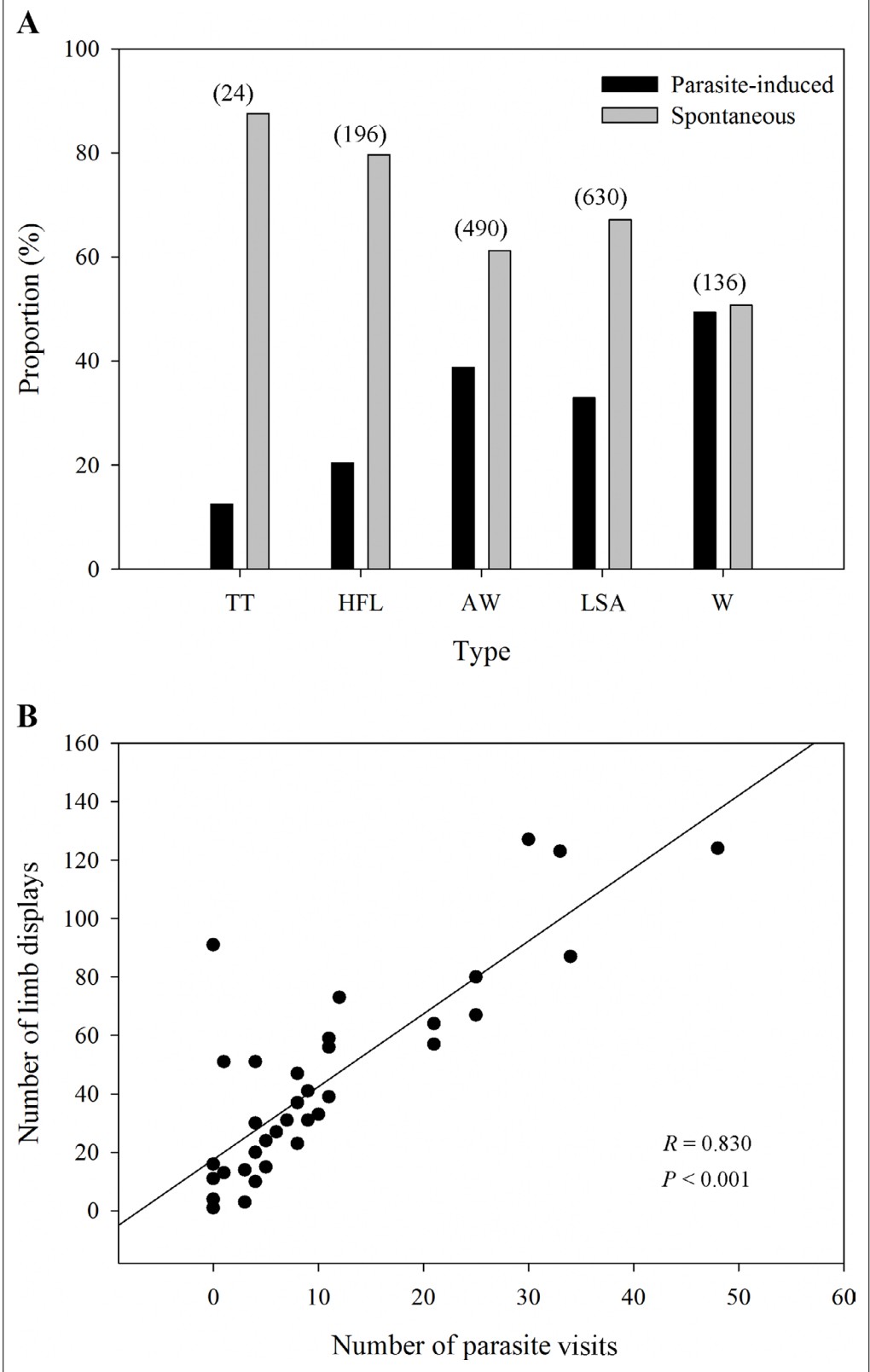

**Figure 2.** The proportion of different limb displays and the correlation between parasites and limb displays. (**A**) The distribution ratio of parasite-induced and spontaneous displays in each limb movement (*N*=69 males). TT, toe trembling; HFL, hind foot lifting; AW, arm waving; LSA, limb shaking; W, wiping. The numbers in brackets above each bar pairs represent the number of each movement, showing the richness of those visual displays.

*Figure 2 continued on next page*

*Figure 2 continued*

(**B**) The relationship between parasite stress and the number of all limb movements (*N*=39 males). Black circles represent different individuals recorded in the field.

p=0.027; *Figure 5B*) movement when compared to a motionless male (static control). Females did not express a preference for the W display (probability = 0.48; *N*=31; p=1; *Figure 5C*) and the LSA display (probability = 0.57; *N*=28; p=0.572; *Figure 5D*) during audio-visual stimulus presentation. Furthermore, females did not express a preference for the AW/HFL stimuli (dynamic visual stimuli) versus motionless stimuli (static visual stimuli) in the absence of an advertisement call (probability = 0.47; *N*=19; p=1). The LS + FF were spontaneous displays (not induced by parasites), and therefore not tested in this experiment. There was a significant association between advertisement call, AW, and HFL displays (Pearson's chi-squared test; *N*=39; $\chi^2_4$=28.18; p<0.001). In particular, the attractive movements (AW and HFL) were strongly associated with advertisement calls (*Figure 6*). The two movements were followed by calls with a probability of 62%, while calls only were followed with movements in 39% of cases.

## Males show more exaggerated displays when females appear nearby

A complete recording of male-female interactions was quite difficult in the wild, because this behavior was rarely observed and frogs frequently moved among stones. Over the past two breeding seasons, we were only able to obtain four recordings of male-female interactions, during which we determined a female was present if it was found within a distance of 1 m to a male (more details are provided in Materials and methods). For those recordings, we only analyzed the LS and FF displays. These data suggest that males use more exaggerated displays (i.e. LS + FF) when females are nearby. We found a higher proportion of males to produce LS and FF displays (the most exaggerated limb movements in little torrent frogs) when females were nearby (Fisher's exact test; $N_1$=4, $N_2$=39; p=0.001; *Figure 7A*). Only 5 out of 39 recorded males emitted LS + FF movements when females or other males were not around (i.e. long-range signaling), while all recorded individuals had such movements when females appeared nearby (i.e. close-range signaling). Moreover, males produced more of those exaggerated displays when females were in close range compared to long-range distance (Wilcoxon rank sum test; $N_1$=4, $N_2$=39; p<0.001; *Figure 7B*).

## Discussion

Male little torrent frogs show a rich repertoire of limb displays. By performing female choice tests and observing male-female interactions in the field, we showed that several of these limb displays (AW, HFL, and LS + FF) can serve as visual signal components and influence female mate choice. All limb movements could be spontaneously generated, but their rates were increased by call-induced parasite presence. We also found calling males to produce more limb displays than silent males. Such parasite-host interactions may have important evolutionary consequences because females in our choice tests showed a strong preference for males that emitted advertisement calls accompanied by parasite-evoked movements (e.g. AW and HFL). We did not find such preference for limb movements

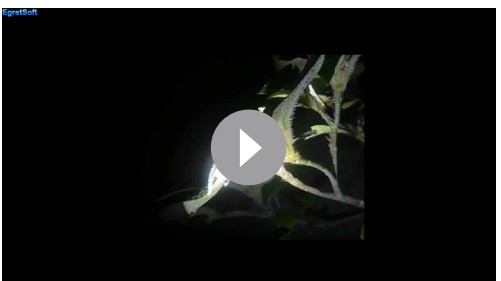

**Video 1.** Video that frog produces defensive motions in order to repel midges.
https://elifesciences.org/articles/76083/figures#video1

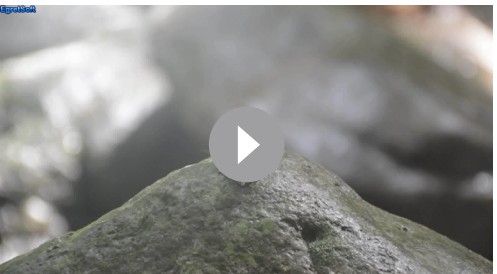

**Video 2.** Video that frog produces a movement toward a flying parasite.
https://elifesciences.org/articles/76083/figures#video2

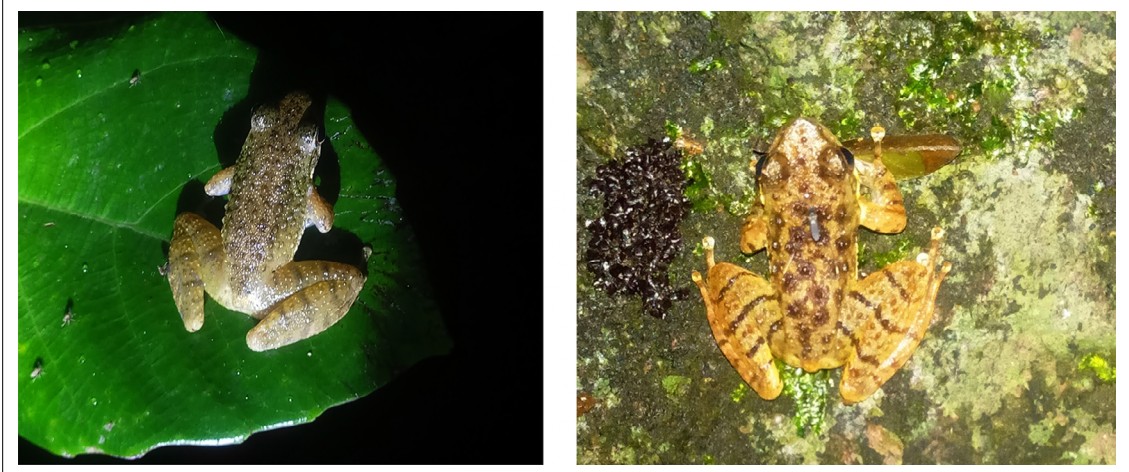

**Figure 3.** Photos of frogs being bitten by potential parasites.

in the absence of call playback, demonstrating that these visual components are part of a multimodal display.

Color, vocal sac, and physical movement, as visual subcomponents, can be added to acoustic signals in anuran multimodal communication (*de Luna et al., 2010*). For example, agonistic behaviors of male *Allobates femoralis* are only evoked when males are exposed to vocal sac pulsations combined with acoustic signals during playback experiments (*Narins et al., 2005*). More and more studies show that streamside-breeding anurans often have complex physical displays including vocal

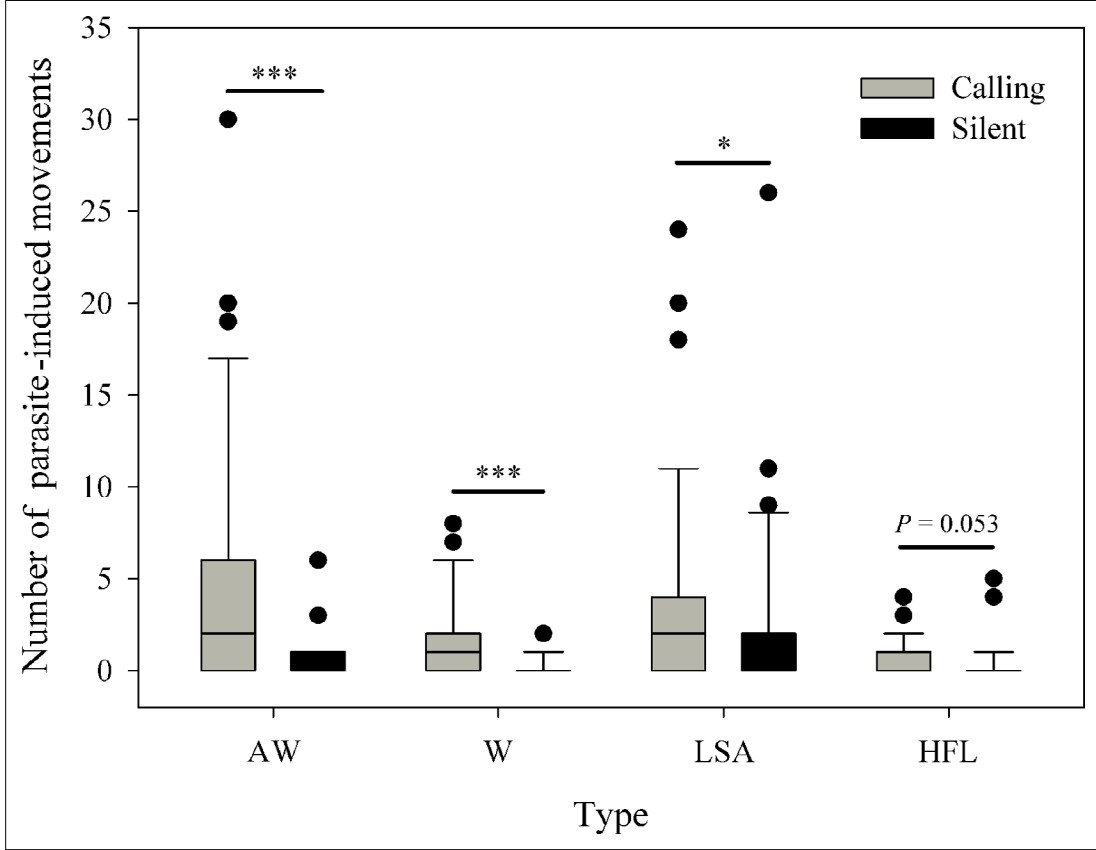

**Figure 4.** Calling males (*N*=39) show more parasite-evoked limb movements than silent males (*N*=30). AW, arm waving; W, wiping; LSA, limb shaking; HFL, hind foot lifting. *p < 0.05, ***p < 0.001. Black circles represent the extreme values of each group.

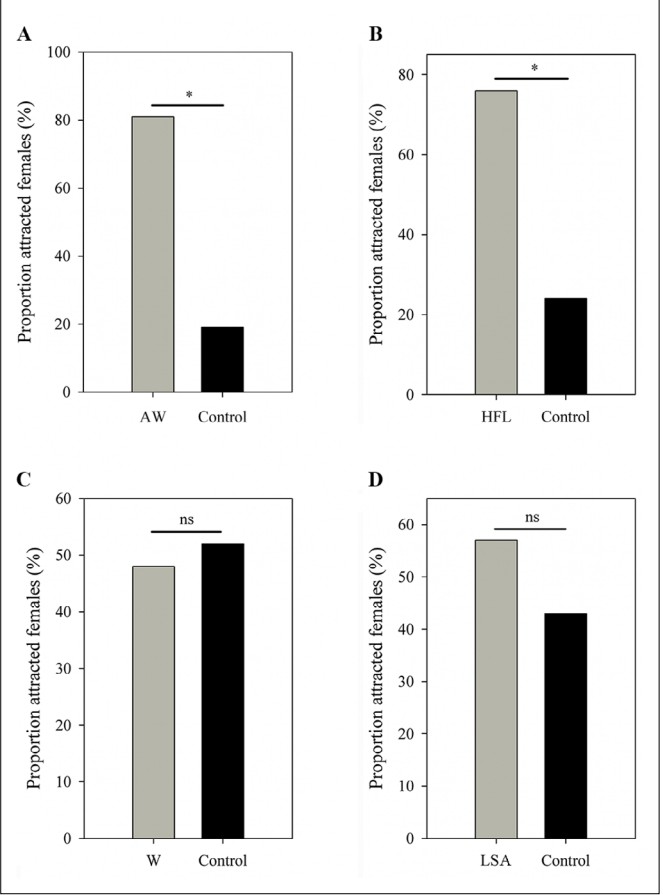

**Figure 5.** Female choices in (**A**) AW versus control, (**B**) HFL versus control, (**C**) W versus control, and (**D**) LSA versus control. All limb display videos are accompanied by advertisement call and male movement, while the controls contain the same call and frog but in absence of limb movement. AW, arm waving; HFL, hind foot lifting; W, wiping; LSA, limb shaking. $^*p < 0.05$. ns, not statistically significant.

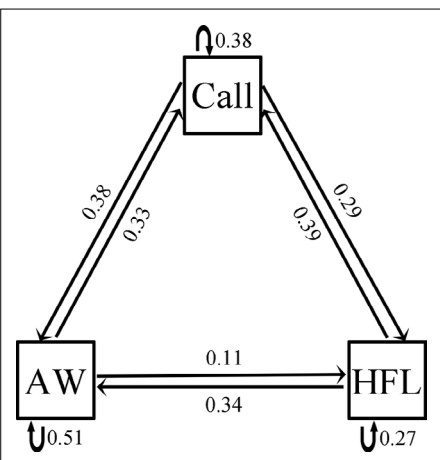

**Figure 6.** Transitional matrix between three behavioral units shown by male little torrent frogs. Call, AW, and HFL represent advertisement call, arm waving, and hind foot lifting, respectively. Numbers next to lines and arrows indicate the transitional probabilities.

sac inflation, limb movement, as well as body movement. For example, a recent investigation found that male Brazilian torrent frogs (*Hylodes japi*) have 18 distinct physical displays which may be associated with different social contexts (*de Sá et al., 2016*). Female little torrent frogs have been shown to prefer the vocal sac movements synchronized with calls over calls only (*Zhao, 2021*). In the present study, we further showed that this species performed diverse common limb displays. However, we did not observe a potential visual signal related with body displays.

Stream-breeding frogs often possess abundant limb or body movements, but their communicative functions have rarely been tested (*Hödl and Amézquita, 2001*). Most previous work analyzed those physical movements based on field observation or video, and few researchers have experimentally explored the effect of these visual displays on female mate choice. In the present study, we experimentally examined whether limb displays can serve as visual signal and influence female choice in little torrent frogs.

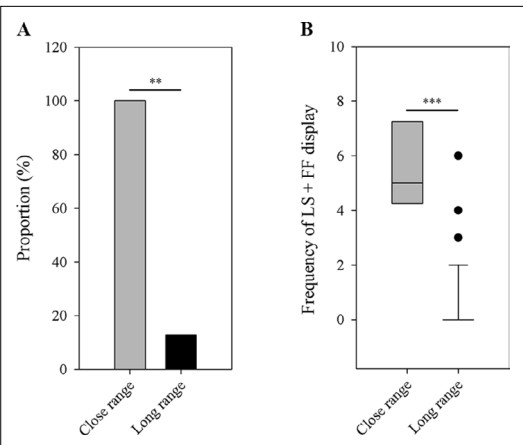

**Figure 7.** Differences between close-range group (*N*=4) and long-range group (*N*=39) in the number of male frogs that have the LS + FF movement (**A**) and the frequency of LS + FF display (**B**). LS, leg stretching; FF, foot flagging. **p < 0.01, ***p < 0.001. Black circles in part B represent the extreme values of long-distance group.

By video playbacks, we found females show a significant preference for males displaying with AW and HFL as compared to motionless males. FF displays are widespread in anurans, and their functions are mainly explored from the perspective of male-male competition. For instance, male Bornean ranid frogs (*Staurois guttatus*) significantly increase such display when conspecific advertisement calls are broadcasted from 1 to 2 m distance in the field (*Grafe and Wanger, 2007*). In our study, males generated more conspicuous LS and FF displays when females were at close range in the field. Because of the difficulties in obtaining male-female interaction data, we only collected four samples which were used to examine the role of LS + FF displays. Similar with results in some other species such as *Staurois parvus* (*Preininger et al., 2013b*), males also increased such movements in response to male-male closed interactions (L Zhao et al., unpublished data), indicating that LS and FF actually are used in close-range communication. Moreover, the statistical power was high and p≤0.001 for all tests. So our results about the role of the exaggerated movements may not be biased by sample size. However, playback tests and more accumulation on field data are necessary in the future.

Frogs and toads are known to be able to see well in dim-light (*Yovanovich et al., 2017*). Many anurans perform behaviors only at environmental illuminations with very low levels (*Buchanan, 1993*). Due to their phototactic responses (*Jaeger and Hailman, 1973*), some video playback studies may have obtained odd results in the past. Little torrent frogs are diurnal species who can communicate with acoustic and visual signals under high-light as well as dark environments. In order to avoid possible effects from light, we set a low environmental illumination according to a natural level. In our study, females discriminated between moving and stationary stimuli. Meanwhile, more conspicuous movements (e.g. HFL) had greater attractiveness than less conspicuous movements (e.g. LSA). We thus argue that little torrent frogs actually recognize the video image as a conspecific stimulus.

Among four visual movements in our playback experiments, the attractive AW and HFL are more conspicuous than the unattractive W and LSA because the display of AW and HFL involves more occupation in terms of time and space. More conspicuous physical movements can increase receiver's attention (*Stokes and Williams, 1971*; *Quaranta et al., 2007*), so the difference in conspicuousness may be related to the attractiveness of these movements. According to the efficacy-based hypotheses, if two signals of different modalities are generated sequentially, a signal may alert the receiver to another signal (*Hebets and Papaj, 2004*; *Grafe et al., 2012*). In this study, the AW and HFL displays tend to be emitted following advertisement calls. We show there is a pattern on the sequence of advertisement calls and limb movements that can elicit female preference, which is similar with the coupling between calls and FF in some torrent frogs (*Preininger et al., 2009*; *Grafe et al., 2012*). The LS and FF movements are the most conspicuous visual signal in little torrent frogs. Males significantly increase the two movements when females appear nearby in the wild. Such conspicuous displays, however, are not used during parasite interactions because male frogs generally perform defensive motions when parasites fall on their body (or limbs) or fly very close to their body (or limbs). Under these conditions, small and medium limb movements are sufficient, while the exaggerated motions (LS and FF) seem to be unnecessary. Therefore, the LS and FF displays are specially performed for conspecific communication, but may have evolved out of the less conspicuous limb movements.

The evolution of dynamic visual signals is often influenced by several social and environmental factors such as territorial aggression (*Preininger et al., 2013a*; *Schuppe et al., 2017*; *Wu et al., 2018*), diurnality (*Harper, 1991*), background noise (*Grafe et al., 2012*; *Grafe and Tony, 2017*), as

well as predation pressure (*Bradbury and Vehrencamp, 1998*). Visual movements or movement-involved multimodal signals can result in senders with more chance of being seen by predators or parasites. Physical displays thus often need to increase the attention of intended receivers while limiting the eavesdropping of unintended receivers (*Steinberg et al., 2014*). In little torrent frogs, male limb displays increased with parasite stress, and such defensive movements can serve as visual cues. These results suggest that parasite stress can induce more visual movements and increase the complexity of audiovisual multimodal displays in little torrent frogs as a by-product. In this study, calling males had a larger parasitic risk than silent males, and we identified *Corethrella* spp. midges which are known to localize frog hosts by eavesdropping on their calls (*Bernal and de Silva, 2015*). However, more studies are needed to further examine whether and how the species is eavesdropped by sound-locating midges.

The idea that some physical movements are the raw material of dynamic visual signals has been proposed for many years (*Harper, 1991*; *Johnsgard, 1962*). In the Panamanian golden frog (*Atelopus zeteki*), for instance, the conspicuous semaphore signal is supposed to originate from a standard stepping motion (*Lindquist and Hetherington, 1998*). However, it is largely unknown how physical movements are incorporated into communicative systems. Sexual selection is believed to be an important driver and the evolution of physical movements may be favored when females are sensitive to some movements in specific environments (*Fleishman, 1992*). During courtship, complex signals are often preferred by females and males frequently include sexual displays more than one channel (*Andersson, 1994*; *Clark and Feo, 2010*). So physical movements from males may be incorporated into multimodal communication systems if they tap into the sensory bias of females and are beneficial to increase the attractiveness of males (sensory exploitation process; *Ryan, 1998*). In our study, female little torrent frogs showed a significant preference for the conspicuous defensive movements when the advertisement calls were simultaneously broadcasted. In noisy streams, acoustic signals plus the relatively conspicuous movements may benefit animals to overcome auditory masking by flowing water (*Partan, 2017*). Our study experimentally shows that such incorporation of non-sexual movements may actually work to increase female preference and thus become part of a multimodal display. The associated limb movements are a by-product of fending of eavesdropping parasites, which may only increase receiver's attention for a brief moment, but which may or may not influence mate choices in other ways (e.g. mate quality assessment). We thus do not argue that we demonstrate a multimodal evolution, but test for prerequisite conditions of a scenario in which multimodal cues evolve from unimodal one via co-option of associated cues.

The simplest movements (LSA/W) are hardly used during parasite interactions and do not induce female preference. The intermediate movements (AW/HFL) are used during parasite interactions and induce a preference. The most complex movements (LS + FF) are only used during male-female (or male-male) interactions. This observed pattern could therefore reflect the evolutionary history of the visual display, from a simple to an advanced stage, where the most complex movements are no longer used in their original context (parasite defense) but only for their new function (sexual communication). Interestingly, in many anurans such as, for example, *Micrixalus saxicola* and *S. parvus* (*Preininger et al., 2013b*), the most complex display also involve FF.

In conclusion, we show that calling behaviors and the levels of parasite interference are correlated and males produce diverse defensive limb movements in order to avoid those unintended receivers in little torrent frogs. By female mate choice tests, we find that relatively conspicuous defensive movements increase the attractiveness of male calls to female frogs. Thus, we suggest that parasite-host interaction may increase the complexity of audiovisual displays. For the first time, we experimentally demonstrate that movements evoked by interspecific activities may evolve via increasing female preference. Our results, together with phylogenetic studies in future, would increase our understanding toward the evolutionary origin of dynamic visual cues and the relationship between natural or sexual selection pressure and multimodal communication behavior.

## Materials and methods
### Field site and study species
The study was carried out in Wuzhishan Nature Reserve (18°55'N and 109°41'E), Hainan Province, China. Average annual rainfall and air temperature in this area are 1800–2000 mm and 22.4°C,

respectively. We focused on the little torrent frog, a species that lives in mountain streams of tropical forests accompanied with high-level torrent noise. Little torrent frogs produce a simple advertisement call (~5.5 s) consisting of a series of repeated notes (~50 notes/call) in which each note (~45 ms) has the dominant frequency around 4 kHz (*Zhao et al., 2018*). Male visual displays, however, are very complex and involve vocal sac inflation and various limb movements. Vocal sac inflation always accompanies call production (fixed composite signal), whereas limb movements can be produced simultaneously with as well as independently from calls (flexibly coupled constituent parts). Likewise, a previous study has shown that the vocal sac inflation does have signal function within multimodal signals (*Zhao, 2021*).

Field data (visual displays and ecological factors) was obtained in May–July 2017 and August–September 2018 in a stream around the management station of Wuzhishan Nature Reserve. Gravid females (characterized by a plump abdomen) were also collected in the stream (from May to July 2019), while female choice tests were conducted in our field lab in Wuzhishan. After being collected, females were brought to the lab in containers which included some water and rocks from their capture sites. Prior to the test, all individuals were placed in a quiet and dark environment for at least 1 hr. In order to avoid repeated testing, individuals were toe-clipped and released at their capture site on the same night after the test. The procedures were approved by the management office of the Wuzhishan Nature Reserve and the Animal Care and Use Committee of the Chengdu Institute of Biology, CAS (CIB2017050004 and CIB2019060012).

## Behavioral recordings and analyses

### Field recordings

We searched for focal males in a stretch of stream (~1.5 km), starting at the station of the Wuzhishan Nature Reserve and ending at the source of the Changhua River. In order to control for differences in temperature, light and daily rhythm, we only recorded calling males versus silent males, between 10 AM and 12 PM (i.e. 2 hr), on sunny days. This species is sexually dimorphic, and females have a larger body size and width-length ratio than males (*Zhao et al., 2017*). We identified them according to those morphological characteristics. Thirty-nine calling frogs and thirty silent frogs were continuously recorded for 10 min using a video camera (GZ-MG465BAC, JVC, Kanagawa, Japan) from a distance of 0.3–0.5 m. The so-called silent frogs did not call at least from the moment of our approach to the end of the recording . A few individuals were excluded because they jumped into the water or started calling during video recordings. We also filmed some males (N=4) that successfully attracted females to close-range and interacted for about 10 min. In the wild, males often significantly decreased calling activity but increased limb displays when conspecific individuals appeared nearby (within 1 m), suggesting that this species may rely more on visual signals in close range (L Zhao et al., unpublished data). Therefore, males with a female close by (within 1 m) and those without one (often without female or other male within 1.5 m) were defined as the close-range and long-range categories, respectively, in this study.

During video recordings, males' calling behaviors and visual displays are not apparently disturbed by our operation, because they always stayed at the original location and performed audio or visual signals as the period prior to be recorded. After each test, the temperature and humidity were measured with an electronic thermohygrometer (YHZ-90450, Yuhuaze, Shenzhen, China) and the background noise (Z-weighted) was measured near the frog's head via a sound level meter (AWA 6291, Hangzhou Aihua Instruments, Hangzhou, China) pointing in vent-snout orientation. We compared the noise levels between calling individuals and silent individuals. The data revealed that the background noise did not differ significantly in two groups (Wilcoxon rank sum test; $N_1$=39, $N_2$=30; $W$=567.5; p=0.837).

### Quantifying limb movements

The limb displays were classified into seven types according to two published ethograms for anurans (see Results). When midges or mosquitoes were on the body of frogs, they would shake their body or move their limbs to repel those parasites (*Video 1*). Frogs also produced limb movements toward parasites that were flying close to their body (*Video 2*). We determined whether or not a limb movement was directly evoked by an insect according to above behaviors. For all frogs, we watched the whole video and scored the number of each spontaneous movement as well as the number of each

display that was evoked by midges or other insects (i.e. insect-induced movement). The number of passive motions was used to represent the level of parasite interference. LS and FF movements were not produced during parasite interactions, and they only had the number of spontaneous displays included. AW and HFL movements can increase the attraction of advertisement calls to female frogs (see Results). We calculated frequencies of the three displays (i.e. call, AW, and HFL) relative to each other, in order to examine if there is a pattern on the sequence of these signals.

## Mate choice tests

### Video recordings and selections

Video stimuli used in our mate choice trials were filmed with a camera (D7100, Nikon, Tokyo, Japan), mounted to a tripod, connected to a directional microphone (MKE400, Sennheiser, Hanover, Germany). The videos were recorded at day in order to obtain better frames, although little torrent frogs can breed and communicate with visual (limb movements) and acoustic signals (calls) day and night. Moreover, they were recorded in front of focal frogs (N=20 males) from a distance of 0.3–0.5 m. We identified four types of limb movement that were often evoked by interactions with parasites. For each of these types (i.e. AW, HFL, W, and LSA), we selected three representative videos, from three different frogs, containing a sequence in which a limb movement was accompanied by a nearby flying midge, and a sequence in which only a flying midge occurred (but with no limb movement). Both sequences were in absence of vocal sac movement. There were two reasons for the exclusion of vocal sac. First, the movement of vocal sac is flexibly coupled with limb displays in natural conditions. Second, the role of limb movements may be masked by vocal sac movement because it has a strong sexual attractiveness and can play a role in mate choice when coupled with advertisement calls (*Zhao, 2021*). TT, LS, and FF displays were not produced during parasite interactions, and thus were not included in the test.

### Stimulus designs

The video stimuli were edited in Adobe Premiere Pro CS6. We firstly cut each clip to 6 s and replicated them to generate a new video with a total length of 10 min, respectively. The display rate of each movement was within a natural range. Next, we changed the audio channel of the video by replacing the original recording with standard sound files. The standard sound files (with flowing water and calls included) were produced according to a stimulus used in a previous study (*Zhao et al., 2017*). The stimulus was synthesized based on the characteristics of 13 males (average dominant frequency: 4.3 kHz) and 3 typical places of flowing water. In little torrent frogs, the call rate of advertisement calls varies from 0.61 to 3.03 calls/min (*Zhao et al., 2018*). In this study, call rate was set to a low level (1 call/min) in order to best simulate conditions in which visual displays might be exploited. For each of four displays, we constructed three audio-visual stimulus pairs (from three different males) always containing a video with a limb display and a video without display for each type of limb movement. In those stimuli, calls were partly overlapped with limb displays on the videos, which were similar with natural scenarios. Moreover, we re-used three videos to produce three stimulus pairs without the advertisement calls (type 5; details are included in Results). Taken together, five types of audio-visual stimuli with a total of 15 pairs were constructed in this experiment (example of each type can be seen in https://www.researchgate.net/publication/359623865).

### Female playbacks

Females were collected and tested at night in order to assure the environmental conditions (e.g. light condition) between behavioral room and field were similar during the experiment. We tested female preferences for our stimulus pairs in a sound-attenuating phonotaxis chamber (1.5 × 1.5 × 1.2 m³, L × W × H) under infrared lighting. We placed a LCD monitor (17S4LSB, Philips, Amsterdam, The Netherlands) and a speaker (JBLCLIP + BLK, JBL, Los Angeles, CA) at each side of two nearby corners, which were used to present sounds and frames, respectively. Such method with frogs in videos has successfully been used to test for sexual preferences in previous studies (*Reichert and Höbel, 2015*; *Rosenthal et al., 2004*). Two monitors were moved and rotated to ensure that both the distance between the two monitors and the distance between the initial female placement point were all 1 m, resulting in a 60° angle between two frames with respect to the initial female placement position (*Figure 8*). All males were life-sized in the screens. The brightness and color of screens were

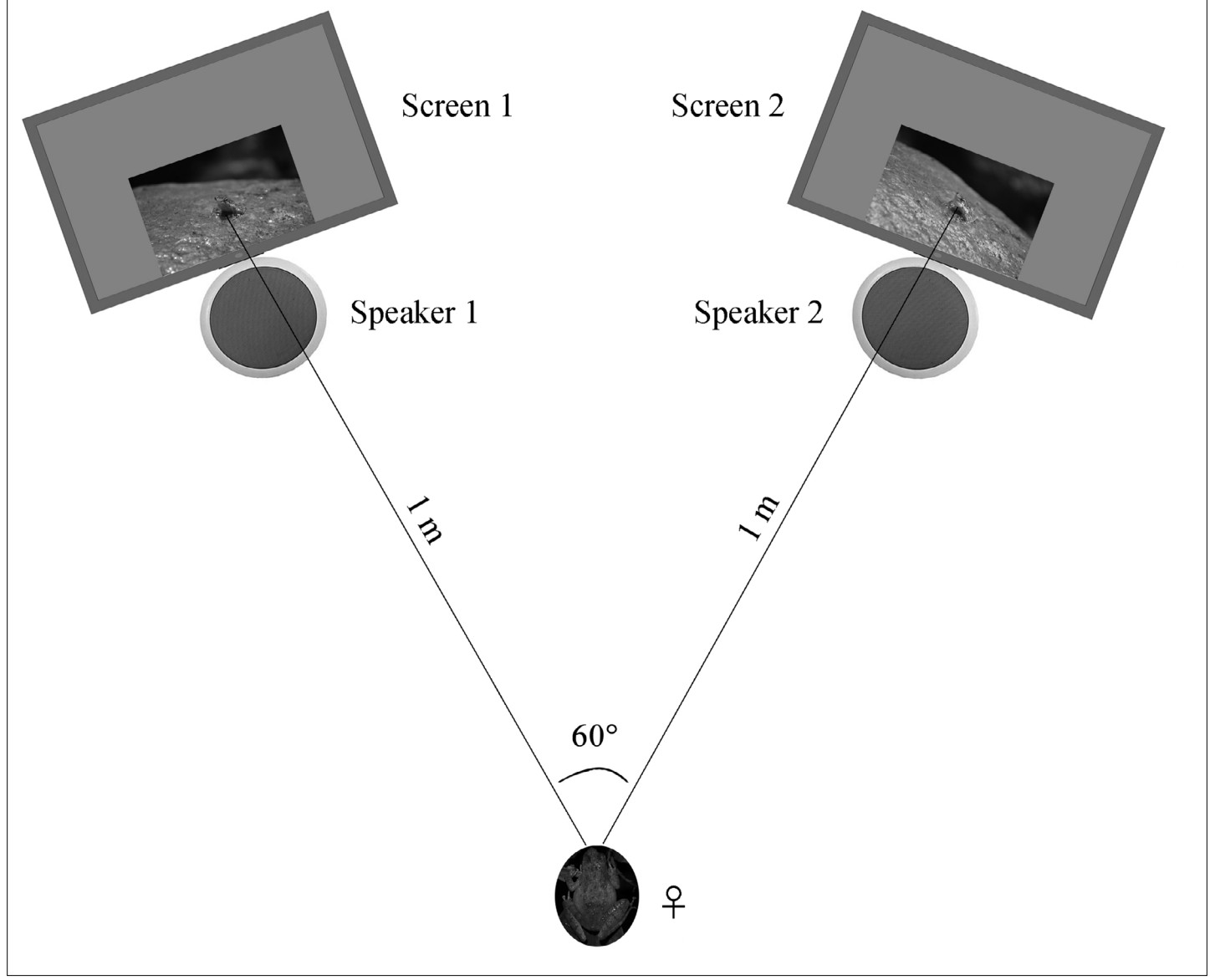

**Figure 8.** Schematic of the acoustic and visual playback arena. The picture of the female frog represents the initial placement point for each playback test.

calibrated using light meter (TES-1399, TES, Taibei, China) and color correction instrument (Spyder5 ELITE, Datacolor, Lawrenceville, GA), respectively. Moreover, the speakers were fixed right under the frame presentation areas during playback (*Figure 8*).

For each female, the sound pressure levels of both speakers were calibrated with a sound level meter such that calls were 75 dB (re 20 µPa) at the initial female placement position. Such intensity is near the auditory threshold (*Zhao et al., 2017*), and was set to increase the likelihood of both calls and movements being noticed (*McDonald et al., 2000*; *Rowe, 1999*). Prior to each playback, a piece of black sponge was placed in front of female frog in order to avoid a possible interference (i.e. light or other visual information prior to each playback) from the screens. The start of each playback was simultaneously conducted with the removing of the sponge during the experiment (females do not move without sound or video playback). A choice was scored when a female approached a speaker-monitor combination within 5 cm. We considered a female as lacking motivation if she failed to make a choice within 10 min. For each frog, we presented the screen displaying a visual stimulus (a moving male) versus the screen displaying the same male but motionless to examine the role of visual displays in absence of acoustic signals. We also presented the screen with audio-visual (call plus movement)

versus the screen with audio (call plus same male not moving) stimulus pairs to test the role of visual displays in presence of acoustic signals. All females thus experienced both movement/no-movement and audio-visual/audio conditions. In order to avoid potential side effects, each stimulus pairs on the left- versus right-sided monitor were randomly broadcasted during the experiments. In order to avoid potential sequence effect, the order of different stimulus pairs was randomized across females. Besides, females could finish multiple tests sequentially and they were not given a break during playback experiments.

## Data analyses

We analyzed all data on male visual display and female mate choice in R (v.3.5.3). We used a Pearson's correlation analysis to determine the relationship between the level of parasite interference and the number of visual display. We carried out Wilcoxon rank sum tests to evaluate the difference of parasite-induced visual displays (i.e. LSA, W, AW, and HFL) between calling frogs and silent frogs (quantity of each per 10 min). We also used the Wilcoxon rank sum test to determine the change of male exaggerated FF and LS displays when a female was close by. In these tests, multiple comparisons were corrected by adjusting p-values using Holm's method. We built a dyadic transition matrix (*Supplementary file 2*) for call, AW, and HFL, and used chi-squared test to determine the association between the three displays (*Preininger et al., 2009*; *Grafe et al., 2012*). We employed Fisher's exact test to examine whether males emitted more exaggerated FF and LS displays when females appeared nearby. Female mate choice was compared with a binomial test. $p < 0.05$ was considered statistically significant.

## Acknowledgements

We thank Yanlin Cai, Xiaoqian Sun, and Xiaofei Zhai for their help during the field recordings. This work was supported by Sichuan Science and Technology Program (2022JDTD0026), National Natural Science Foundation of China (31772464), Youth Innovation Promotion Association CAS (2012274), and CAS 'Light of West China' Program.

## Additional information

### Funding

| Funder | Grant reference number | Author |
|---|---|---|
| Sichuan Science and Technology Program | 2022JDTD0026 | Jianguo Cui |
| National Natural Science Foundation of China | 31772464 | Jianguo Cui |
| Youth Innovation Promotion Association | 2012274 | Jianguo Cui |
| CAS "Light of West China" Program | | Jianguo Cui |

The funders had no role in study design, data collection and interpretation, or the decision to submit the work for publication.

### Author contributions

Longhui Zhao, Data curation, Formal analysis, Methodology, Writing – original draft, Writing – review and editing; Jichao Wang, Haodi Zhang, Data curation, Investigation; Tongliang Wang, Data curation, Resources; Yue Yang, Investigation, Methodology; Yezhong Tang, Methodology, Writing – original draft; Wouter Halfwerk, Writing – original draft, Writing – review and editing; Jianguo Cui, Conceptualization, Funding acquisition, Methodology, Writing – original draft, Writing – review and editing

### Author ORCIDs

Longhui Zhao http://orcid.org/0000-0001-8746-2803

### Ethics

All procedures were approved by the management office of the Wuzhishan Nature Reserve and the Animal Care and Use Committee of the Chengdu Institute of Biology, CAS (CIB2017050004 & CIB2019060012).

### Decision letter and Author response

Decision letter https://doi.org/10.7554/eLife.76083.sa1
Author response https://doi.org/10.7554/eLife.76083.sa2

## Additional files

### Supplementary files

• Supplementary file 1. Table S1. The data of total (spontaneous and parasite-induced) and parasite-induced displays in each limb movement for calling males, silent males, and males that have females nearby.

• Supplementary file 2. Table S2. Dyadic transition matrix of three behavioral units.

• Transparent reporting form

### Data availability

Data used to generate the results are available from the Dryad Digital Repository: https://doi.org/10.5061/dryad.f1vhhmgzg.

The following dataset was generated:

| Author(s) | Year | Dataset title | Dataset URL | Database and Identifier |
|---|---|---|---|---|
| Zhao L, Wang J, Zhang H, Wang T, Yang Y, Tang Y, Halfwerk W, Cui J | 2022 | The data of parasite-induced and spontaneous displays in each limb movement for calling males, silent males and males that have females nearby | https://dx.doi.org/10.5061/dryad.f1vhhmgzg | Dryad Digital Repository, 10.5061/dryad.f1vhhmgzg |

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
