## [Editor Report]

Zhao et al., present an intriguing proposal for the evolution of complex multimodal signals based on the analysis of both acoustic and visual signals of small torrent frogs’ mating displays. Combining field observations with experiments, they suggest that male limb movements, which are used to swat away blood-sucking midges, have become attractive to female frogs, demonstrating how these movements enhance the mating calls of males. Their research suggests a potential pathway for hirtyle-host interactions to become co-opted into sexually selected mating displays.

---

## [Decision Letter]

**Decision letter after peer review:**

Thank you for submitting your article “Parasite defensive limb movements enhance signal attraction in male little torrent frogs: insight into the evolution of multimodal signals” for consideration by *eLife*. Your article has been reviewed by two peer reviewers, and the evaluation has been overseen by Ammie Kalan as the Reviewing Editor and Christian Rutz as the Senior Editor. The following individual involved in the review of your submission has agreed to reveal their identity: Iris Starnberger (Reviewer #1).

The reviewers have discussed their reviews with one another, and the Reviewing Editor has drafted this decision letter to help you prepare a revised submission.

Essential revisions:

Both reviewers found the manuscript to be a valuable contribution to understanding the evolution of multimodal, complex signals, including the need for adding relevant information to the introduction about display by-products and their role in signal enhancement, which will help to contextualize the study's findings (see detailed reviewer reports below). That said, we feel an additional analysis should be added that could significantly strengthen the work: could you please investigate whether there is a sequenced, repetitive pattern to the limb movements of the males that might be considered a visual display? We think that even if it turns out that there is no pattern, this result would be important to include in the manuscript.

*Reviewer #1 (Recommendations for the authors):*

Abstract:

Line 32: “(…) which combine multiple programs.” – I am not sure, what you mean by “programs”. I would suggest changing to “(…) which combine multiple modalitites.”

Line 33: “(…) ecological and sexual selection (…)” – I would suggest changing to “(…) natural and sexual selection (…)”, if the authors agree.

Introduction:

Line 46: “(…) or increased of predation (…)” – This sentence is confusing to me. It seems, like a word may be missing – please rephrase.

Line 63: “(…) intention movements (…)” I would suggest changing to “intentional movements”

Line 66: “Anurans (i.e. frogs) provide (…)” I would suggest changing to “Anurans (i.e. frogs and toads) provide (…)”

Line 80: “(…) movements are often similar with limb movements (…)” – change to “(…) movements are often similar to limb movements (…)”

Line 91: “(…) generate limb movements (…)” – change to „(…) display limb movements (…)”

Line 92: “(…) to visual displays that have been reported in other torrent frogs.” – I would suggest adding references.

Line 99: “Second, we assessed (…)” – “Secondly, we assessed (…)”

Line 101: “(…) and tested whether calling males produced more parasite-evoked displays compared to silent males.” It might be beneficial to add a quick explanation here to clarify how you know that you are not testing individual differences in the reaction to the midges, but rather the effects of the midges on the display behaviour overall.

Results:

General comment: The structure with the subtitles really improved readability. However, for me personally it was a bit of a struggle to remember what all the abbreviatons (e.g. HFL, TT, LS, etc.) referred to. It might serve the reader better if you either omit the abbreviations altogether, or if you remind the reader of their meaning in each of the manuscript’s sub-sections.

Line 123: “(…) categorized this results (…)” – change to “(…) categorized these results (…)”

Line 129: “(…) Corethrella spp (…)” – change to “(…) Corethrella spp. (…)”

Lines 168-182: How was it determined that a female was present? What was the difference between a female being in close range versus at a long range distance? I do realize that this information belongs into the Methods section, but that question kept popping up in my head while reading the Results section.

Discussion:

Line 184: “Male little torrent frogs (…)” – “Male small torrent frogs (…)”

Line 212: “(…) significantly increases (…)” – change to “(…) significantly increase (…)”

Line 235: “(…) as well as darkness environments.” – change to „(…) as well as dark environments (…)”

Line 267: “(…) Corethrella spp (…)” – change to “(…) Corethrella spp. (…)”

Line 282: “(…) females little torrent frogs (…)” – change to “(…) female little torrent frogs (…)”

Line 285: “(…) plus the relative conspicuous (…)” – change to “(…) plus the relatively conspicuous (…)”

Line 301: “(…) we find the relative conspicuous defensive movements (…)” – change to “(…) we find that relatively conspicuous defensive movements (…)”

Methods:

General comment: While I applaud the notion of using subtitles to improve readability, to me the sections per sub title seemed rather long and I would suggest adding further subsections to clarify the structure. Furthermore, I would suggest thoroughly re-reading and potentially rephrasing this section, as to me the Methods section seemed to be a bit „sketchy”, which was in strong contrast to the rest of the manuscript, and made following the authors' thoughts a bit difficult.

Line 319: “(…) as well as other physical movements (…)” Which ones? Could you give a few examples for other body movements?

Line 320: “(obligatory coupled signal components)” – if the authors agree, I would suggest changing to “(fixed composite signal)”

Line 320: “(…) can be made simultaneous as well as independent from calls (…)” – change to “(…) can be produced simultaneously with as well as independently from calls (…)”

Line 322: “(…) a previous study has showed that (…)” – change to “(…) a previous study has shown that (…)” and maybe change to “that the vocal sac inflation does have signal function within multimodal signals.”

Lines 324-339: This is one of the sections that seemed quite confusing to me.

Lines 345-346: “(…) females have a larger body size and width-length ratio than males.” Please add a reference to this statement.

Line 347: “Thirty-nine calling frogs and thirty silent frogs were continuously recorded (…)”. How were the silent frogs found? Were they calling before? Did some of the recordings have to be discarded, because the silent focal males started calling?

Line 398: “(…) according to an stimulus (…)” – change to „(…) according to a stimulus (…)”

Line 413: “(…) placed a LCD monitor (…)” – change to “(…) placed an LCD monitor (…)”

Line 435: “(…) we presented the screen with a visual movement (a moved male) versus the screen without the visual movement (…)”. I would suggest changing to “(…) we presented the screen displaying a conspicuous visual stimulus (a moving male) versus the screen displaying the same male, but motionless (…)”

Line 440: How many females were tested under these conditions?

Line 455: “(…) displays when females was around.” – change to “(…) displays when a female was close by.”

Acknowledgments:

Line 461: “(…) help during the wild recordings (…)” I would suggest changing to “(…) help during the field recordings (…)”

Figure 1: “(…) The picture of female frog (…)” change to “(…) the picture of the female frog (…)”

Figure 2B: Y-Axis – change to “displays”; X-Axis – change to “visits”

Figure 3: Y-Axis – change to “movements”

Figure 4: Line 21: “(…) but in absence of movement (…)” – Was the vocal sac movement (corresponding to the advertisement call) visible? If yes, I would probably rephrase to “(…) but in absence of limb movement (…)”.

Supplementary 1: I really enjoy Figure S1 and would suggest moving it into the main part of the manuscript. And – really just a friendly suggestion – some of these illustrations could be added to e.g. Figure 3 or other figures. That would make the graphs very self-explanatory and even more aesthetically pleasing.

Figure S2: “Photos that frogs are being bitten (…)” – change to “Photos of frogs being bitten (…)”

I would suggest moving the two photos to the main part of the manuscript.

*Reviewer #2 (Recommendations for the authors):*

Lines 54-65: This section needs more work. Please expand on the connection of a stereotyped signal (e.g., mating call) and by-products of other behaviors (e.g., antiparasitic or antipredator movements). I am not convinced that chaotic limb movements by males are necessarily a part of a multimodal mating display. Instead, limb movements have the effect of priming the attention of females (i.e., they attract the attention of the receiver) before the actual mating signal is transmitted to its intended receiver. There are many reports on this subject that the authors might like to review and summarize in their introduction:

Clark, David L., and Carrie L. Morjan. “Attracting Female Attention: The Evolution of Dimorphic Courtship Displays in the Jumping Spider Maevia Inclemens (Araneae: Salticidae).” Proceedings: Biological Sciences, vol. 268, no. 1484, The Royal Society, 2001, pp. 2461-65

Hugill, N., Fink, B., and Neave, N. (2010). The role of human body movements in mate selection. Evolutionary psychology 8(1), 66-89

Lines 78-84: The closest study to the one presented in this manuscript, I think, is by Walter Hold’s research team on several ranid frogs. It would be important to summarize this research as the manuscript refers to such earlier studies in many parts of the discussion.

Preininger, D., Boeckle, M., Hold, W. 2009. Communication in Noisy Environments II: Visual Signaling Behavior of Male Foot-flagging Frogs staurois Latopalmatus. Herpetologica 65(2):166-173

Lines 109-124: I am not sure if limb movements to swat or repel parasitic midges should be considered as visual displays. In anurans, mating displays are mostly stereotyped and tend to have a rather fixed and predictable sequence of components. How do the authors distinguish those that represent “scratching reflexes” (e.g., HFL, AW, LSA, W) from a highly stereotyped mating call? Should they consider any body movement as a visual display? What are those body movements that seem to be true visual displays (i.e., an intentioned movement to attract a mate over to repel a parasite)?

Lines 156-167: This paragraph is probably the novelty of this research. The authors report that females prefer males that show some form of body movement versus those that are motionless if such males are vocalizing. Again, for me, the results do support those moving males are preferred over the motionless, yet I am not sure that limb movement is part of the mating display.

My suggestion is to review the videos and determine if there is a pattern on the sequence of limb movements that elicits female preference. If such analyses reveal that a specific sequence of movement is preferred by females, then this provides stronger evidence for a multimodal mating signal. If there is not a pattern, I would consider that moving males become more attractive by capturing more easily females’ attention. In contrast, if females respond to a specific sequence of body movements, this is better evidence that mating displays are multimodal.

Lines 168-183: The results might have been of relevance, yet the tiny sample size of just N1=4 might prevent any sort of robust inference. In other words, I am not sure that males with more exaggerated movements are more attractive than those that moved less. The sample size is too small.

Supporting data: The authors should make the videos associated with this study available to the public as they will provide a nice teaching tool for students.

Overall, I consider this manuscript a nice natural history report; yet some of the interpretations might not be supported with the evidence at hand. First, I am not sure that limb movements without a pattern could be a component of a multimodal mating signal. Such movements might attract female attention onto a focal male, yet this individual must vocalize. Second, higher levels of body movement by the males that make them more attractive might not be supported by the low number of observations used in the statistical analyses. This interpretation requires further fieldwork.

---

## [Author Response]

Essential revisions:Both reviewers found the manuscript to be a valuable contribution to understanding the evolution of multimodal, complex signals, including the need for adding relevant information to the introduction about display by-products and their role in signal enhancement, which will help to contextualize the study’s findings (see detailed reviewer reports below). That said, we feel an additional analysis should be added that could significantly strengthen the work: could you please investigate whether there is a sequenced, repetitive pattern to the limb movements of the males that might be considered a visual display? We think that even if it turns out that there is no pattern, this result would be important to include in the manuscript.

We thank you for the useful and interesting suggestions to add additional relevant information to our introduction and to re-analyse some of our data.

(1) We have added the information about display by-products in the introduction according to reviewer reports below. Please see revised p. 3-4 lines 53-75.

(2) We have built a dyadic transition matrix (table S2) for three behavioral units (i.e. advertisement call, AW, and HFL), and determined the associations between the three displays using a method from Preininger et al., (2009) and Grafe et al., (2012). Interestingly, we found that the AW and HFL displays tended to be emitted following advertisement calls. We thus show there is a pattern on the sequence of advertisement calls and limb movements that can elicit female preference. Such call-mediated pattern can provide a chance of multimodal (acoustic and visual) communication. We have added the relevant information in introduction (the revised p. 6-7 lines 126-127), results (the revised p. 9-10 lines 191-195), discussion (the revised p. 13-14 lines 276-280), and methods (the revised p. 20 lines 415-419 and p. 24 lines 510-511).

Reviewer #1 (Recommendations for the authors):Abstract:Line 32: "(…) which combine multiple programs." – I am not sure, what you mean by "programs". I would suggest changing to "(…) which combine multiple modalitites."

Now changed. Please see revised p. 2 line 32.

Line 33: "(…) ecological and sexual selection (…)" – I would suggest changing to "(…) natural and sexual selection (…)", if the authors agree.

Now changed. Please see revised p. 2 line 33.

Introduction:Line 46: "(…) or increased of predation (…)" – This sentence is confusing to me. It seems, like a word may be missing -- please rephrase.

We have rephrased the sentence. Please see revised p. 3 lines 44-46.

Line 63: "(…) intention movements (…)" I would suggest changing to "intentional movements"

This was done. Please see revised p. 4 line 68.

Line 66: "Anurans (i.e. frogs) provide (…)" I would suggest changing to "Anurans (i.e. frogs and toads) provide (…)"

Now included. Please see revised p. 4 line 76.

Line 80: "(…) movements are often similar with limb movements (…)" – change to "(…) movements are often similar to limb movements (…)"

Now corrected. Please see revised p. 5 lines 94-95.

Line 91: "(…) generate limb movements (…)" – change to „(…) display limb movements (…)"

Now changed. Please see revised p. 6 line 108.

Line 92: "(…) to visual displays that have been reported in other torrent frogs." – I would suggest adding references.

We have included two representative references in the text. Please see revised p. 6 line 109.

Line 99: "Second, we assessed (…)" – "Secondly, we assessed (…)"

Now corrected. Please see revised p. 6 line 119.

Line 101: "(…) and tested whether calling males produced more parasite-evoked displays compared to silent males." It might be beneficial to add a quick explanation here to clarify how you know that you are not testing individual differences in the reaction to the midges, but rather the effects of the midges on the display behaviour overall.

In order to avoid individual differences, we collected sufficient samples (at least 30 males in each group) to compare the defend displays between calling frogs and silent frogs inhabited in same environment. We have clarified this in the text. Please see revised p. 6 lines 122-124.

Results:General comment: The structure with the subtitles really improved readability. However, for me personally it was a bit of a struggle to remember what all the abbreviatons (e.g. HFL, TT, LS, etc.) referred to. It might serve the reader better if you either omit the abbreviations altogether, or if you remind the reader of their meaning in each of the manuscript's sub-sections.

Thank you for your good advices. Now we have reminded the meaning of these abbreviations in all sub-sections of the manuscript (e.g. the revised p. 8 lines 155-158).

Line 123: "(…) categorized this results (…)" – change to "(…) categorized these results (…)"

We are sorry for the mistake. Now corrected. Please see revised p. 7 line 146.

Line 129: "(…) Corethrella spp (…)" – change to "(…) Corethrella spp. (…)"

Now changed. Please see revised p. 8 line 152.

Lines 168-182: How was it determined that a female was present? What was the difference between a female being in close range versus at a long range distance? I do realize that this information belongs into the Methods section, but that question kept popping up in my head while reading the Results section.

We determined a female was present if it was attracted it to a distance within 1 m by a male. In the wild, males often significantly decreased calling activity but increased limb displays when conspecific individuals appeared nearby (within 1 m), suggesting that this species may rely more on visual signals in close range (L. Zhao et al., unpublished data). Therefore, males with a female close by (within 1 m) and those without one (often without female or other male within 1.5 m) were defined as the close-range and long-range categories, respectively. We have clarified them in the revised p. 10 lines 200-201 and p. 18-19 lines 387-392.

Discussion:Line 184: "Male little torrent frogs (…)" – "Male small torrent frogs (…)"

In published papers and books (e.g. Fei et al., 2012; Preininger et al., 2013), small torrent frogs refer to *Micrixalus saxicola*, while little torrent frogs refer to *Amolops torrentis*. Therefore, "Male little torrent frogs (…)" is correct.

Line 212: "(…) significantly increases (…)"– change to "(…) significantly increase (…)"

This was done. Please see revised p. 12 lines 247-248.

Line 235: "(…) as well as darkness environments." – change to „(…) as well as dark environments (…)"

Now changed. Please see revised p. 13 line 263.

Line 267: "(…) Corethrella spp (…)" – change to "(…) Corethrella spp. (…)"

Thank you. We have corrected this in the text. Please see revised p. 14 line 299.

Line 282: "(…) females little torrent frogs (…)" – change to "(…) female little torrent frogs (…)"

Now changed. Please see revised p. 15 lines 314-315.

Line 285: "(…) plus the relative conspicuous (…)" – change to "(…) plus the relatively conspicuous (…)"

We have corrected this in the text. Please see revised p. 15 line 317.

Line 301: "(…) we find the relative conspicuous defensive movements (…)" – change to "(…) we find that relatively conspicuous defensive movements (…)"

Now changed. Please see revised p. 16 lines 338-339.

Methods:General comment: While I applaud the notion of using subtitles to improve readibility, to me the sections per sub title seemed rather long and I would suggest adding further subsections to clarify the structure. Furthermore, I would suggest thoroughly re-reading and potentially rephrasing this section, as to me the Methods section seemed to be a bit „sketchy", which was in strong contrast to the rest of the manuscript, and made following the authors' thoughts a bit difficult.

Thanks for your good suggestions. We have added further subsections to clarify the structure. We also re-read and improve the whole section. Please see revised p. 17-24 lines 347-514.

Line 319: "(…) as well as other physical movements (…)" Which ones? Could you give a few examples for other body movements?

We now removed these words because “the physical movements” is repetitive with previous “limb movements”.

Line 320: "(obligatory coupled signal components)" – if the authors agree, I would suggest changing to "(fixed composite signal)"

We agree with you and have changed this in the text. Please see revised p. 17 line 357.

Line 320: "(…) can be made simultaneous as well as independent from calls (…)" – change to "(…) can be produced simultaneously with as well as independently from calls (…)"

Now changed. Please see revised p. 17 line 358.

Line 322: "(…) a previous study has showed that (…)" – change to "(…) a previous study has shown that (…)" and maybe change to "that the vocal sac inflation does have signal function within multimodal signals."

We are sorry for the mistake. Now corrected. Please see revised p. 17 line 359.

Lines 324-339: This is one of the sections that seemed quite confusing to me.

We have refined and improved this paragraph. Please see revised p. 17-18 lines 361-372.

Lines 345-346: "(…) females have a larger body size and width-length ratio than males." Please add a reference to this statement.

Now added. Please see revised p. 18 line 380.

Line 347: "Thirty-nine calling frogs and thirty silent frogs were continuously recorded (…)". How were the silent frogs found? Were they calling before? Did some of the recordings have to be discarded, because the silent focal males started calling?

This species prefer to crouch on stones around fast-flowing water. We searched for silent males according to our field experience. The silent frogs did not calling at least from our approach moment to final recording moment. A few individuals were discarded because they jumped to water or started calling during video recordings. We have clarified them in the text. Please see revised p. 18 lines 383-385.

Line 398: "(…) according to an stimulus (…)" – change to „(…) according to a stimulus (…)"

Now changed. Please see revised p. 21 line 447.

Line 413: "(…) placed a LCD monitor (…)" -> change to "(…) placed an LCD monitor (…)"

Now changed. Please see revised p. 22 line 466.

Line 435: "(…) we presented the screen with a visual movement (a moved male) versus the screen without the visual movement (…)". I would suggest changing to "(…) we presented the screen displaying a conspicuous visual stimulus (a moving male) versus the screen displaying the same male, but motionless (…)"

This was done. Please see revised p. 23 lines 488-490.

Line 440: How many females were tested under these conditions?

The number of females was different among different stimulus pairs. We have provided the number of each group in Results. Please see revised p. 9 lines 181-185.

Line 455: "(…) displays when females was around." – change to "(…) displays when a female was close by."

Now changed. Please see revised p. 24 line 508.

Acknowledgments:Line 461: "(…) help during the wild recordings (…)" I would suggest changing to "(…) help during the field recordings (…)"

Now changed. Please see revised p. 24 lines 516-517.

Figure 1: "(…) The picture of female frog (…)" change to "(…) the picture of the female frog (…)"

Now included. Please see revised Figure 8 legends (p. 34 line 711).

Figure 2B: Y-Axis – change to "displays"; X-Axis – change to "visits"

Now changed. Please see revised Figure 2.

Figure 3: Y-Axis – change to "movements"

Now changed. Please see revised Figure 4.

Figure 4: Line 21: "(…) but in absence of movement (…)" – Was the vocal sac movement (corresponding to the advertisement call) visible? If yes, I would probably rephrase to "(…) but in absence of limb movement (…)".

The vocal sac was not visible in all stimulus pairs. We have changed it to "(…) but in absence of limb movement (…)" in order to avoid confusing readers. Please see revised Figure 5 legends (p. 33 line 701).

Supplementary 1: I really enjoy Figure S1 and would suggest moving it into the main part of the manuscript. And -- really just a friendly suggestion -- some of these illustrations could be added to e.g. Figure 3 or other figures. That would make the graphs very self-explanatory and even more aesthetically pleasing.

(1) We have moved the Figure S1 into the main manuscript. Please see revised Figure 1 and p. 7 lines 135-144.

(2) We have tried to place the diagrams into Figures, but some displays (e.g. toe trembling) are not visible after being zoomed out.

Figure S2: "Photos that frogs are being bitten (…)" – change to "Photos of frogs being bitten (…)"I would suggest moving the two photos to the main part of the manuscript.

We have changed this and moved the two photos to the manuscript. Please see revised Figure 3 and p. 33 line 694.

Reviewer #2 (Recommendations for the authors):Lines 54-65: This section needs more work. Please expand on the connection of a stereotyped signal (e.g., mating call) and by-products of other behaviors (e.g., antiparasitic or antipredator movements). I am not convinced that chaotic limb movements by males are necessarily a part of a multimodal mating display. Instead, limb movements have the effect of priming the attention of females (i.e., they attract the attention of the receiver) before the actual mating signal is transmitted to its intended receiver. There are many reports on this subject that the authors might like to review and summarize in their introduction:Clark, David L., and Carrie L. Morjan. "Attracting Female Attention: The Evolution of Dimorphic Courtship Displays in the Jumping Spider Maevia Inclemens (Araneae: Salticidae)." Proceedings: Biological Sciences, vol. 268, no. 1484, The Royal Society, 2001, pp. 2461-65Hugill, N., Fink, B., and Neave, N. (2010). The role of human body movements in mate selection. Evolutionary psychology 8(1), 66-89

(1) Thank you. We have included more reports about this subject in the introduction. Please see revised p. 3-4 lines 53-75.

(2) Regarding the chaotic limb movements, we would predict an association in time between calling and limb-movements, as the acoustic component of the call attracts parasites, which induces several types of limb movements. In response to your comments and suggestions, we now include additional analyses about signal sequence and shown that limb movements are indeed associated with calling behaviour and therefore not random. Consequently, these limb movements, which are a by-product of calling-induced parasitism, have the potential to become incorporated as part of a multimodal display.

Lines 78-84: The closest study to the one presented in this manuscript, I think, is by Walter Hold's research team on several ranid frogs. It would be important to summarize this research as the manuscript refers to such earlier studies in many parts of the discussion.Preininger, D., Boeckle, M., Hold, W. 2009. Communication in Noisy Environments II: Visual Signaling Behavior of Male Foot-flagging Frogs staurois Latopalmatus. Herpetologica 65(2):166-173

This was done now. Please see revised p. 5 lines 88-94.

Lines 109-124: I am not sure if limb movements to swat or repel parasitic midges should be considered as visual displays. In anurans, mating displays are mostly stereotyped and tend to have a rather fixed and predictable sequence of components. How do the authors distinguish those that represent "scratching reflexes" (e.g., HFL, AW, LSA, W) from a highly stereotyped mating call? Should they consider any body movement as a visual display? What are those body movements that seem to be true visual displays (i.e., an intentioned movement to attract a mate over to repel a parasite)?

(1) We think some midge-evoked movements (i.e. AW and HFL) should be considered as visual displays because they can increase the attraction of male calls to female frogs (please see revised p. 9 lines 180-183). Moreover, our analyses on transitional matrix between three behavioral units showed a significant association between advertisement call, AW, and HFL displays. In particular, the attractive movements (i.e. AW and HFL) were strongly associated with advertisement calls. The two movements followed calls with the probability of 62%, while calls only were followed with movements 39% of the time (please see revised p. 9-10 lines 191-195). So there is a specific sequence between advertisement calls and limb movements that can elicit female preference. More interestingly, the pattern is similar with the coupling between calls and FF in some other torrent frogs. We have discussed this in the manuscript. Please see revised p. 13-14 lines 276-280.

(2) The mating call and defensive movement are different modalities (acoustic signal versus visual signal) and can be easily distinguished in video recordings. According to our present results, FF and LS are intentioned movements which are used to conspecific communication, while other movements seem to be emitted to repel parasites.

Lines 156-167: This paragraph is probably the novelty of this research. The authors report that females prefer males that show some form of body movement versus those that are motionless if such males are vocalizing. Again, for me, the results do support those moving males are preferred over the motionless, yet I am not sure that limb movement is part of the mating display.My suggestion is to review the videos and determine if there is a pattern on the sequence of limb movements that elicits female preference. If such analyses reveal that a specific sequence of movement is preferred by females, then this provides stronger evidence for a multimodal mating signal. If there is not a pattern, I would consider that moving males become more attractive by capturing more easily females' attention. In contrast, if females respond to a specific sequence of body movements, this is better evidence that mating displays are multimodal.

Thank you for your great suggestion. We have reviewed 39 calling videos and built a dyadic transition matrix (table S2) for three behavioral units (i.e. advertisement call, AW, and HFL). Meanwhile, we determined the associations between the three displays using a method from Preininger et al., (2009) and Grafe et al., (2012). As a result, we found the AW and HFL displays tended to be emitted following advertisement calls. Therefore, there is a pattern on the sequence of advertisement calls and limb movements that can elicit female preference. These additional analyses provide a strong evidence for multimodal mating signals. We have included the relevant information in the manuscript. Please see revised p. 6-7 lines 126-127, p. 9-10 lines 191-195, p. 13-14 lines 276-280, p. 20 lines 415-419, and p. 24 lines 510-511.

Lines 168-183: The results might have been of relevance, yet the tiny sample size of just N1=4 might prevent any sort of robust inference. In other words, I am not sure that males with more exaggerated movements are more attractive than those that moved less. The sample size is too small.

Because of the difficulties in obtaining male-female interaction data, we only collected four samples which were used to examine the role of LS + FF displays. Similar with results in some other species such as *Staurois parvus* (Preininger et al., 2013), males also increased such movements in response to male-male closed interactions (L. Zhao et al., unpublished data), indicating that LS and FF actually are used in close-range communication. Moreover, the statistical power was high and *P* ≤ 0.001 for all tests. So our results about the role of the exaggerated movements may not be biased by sample size. However, playback tests and more accumulation on field data are necessary in the future. We have clarified them in the text. Please see revised p. 12-13 lines 250-258.

Supporting data: The authors should make the videos associated with this study available to the public as they will provide a nice teaching tool for students.

Thank you for your good advice. We have shared these videos in a public web. Please see revised p. 22 lines 459-460.

Overall, I consider this manuscript a nice natural history report; yet some of the interpretations might not be supported with the evidence at hand. First, I am not sure that limb movements without a pattern could be a component of a multimodal mating signal. Such movements might attract female attention onto a focal male, yet this individual must vocalize. Second, higher levels of body movement by the males that make them more attractive might not be supported by the low number of observations used in the statistical analyses. This interpretation requires further fieldwork.

Thank you for your insightful comments again. We have addressed the two concerns according to previous comments. Please see the responses to the comments above.